# A pragmatic randomised controlled trial of 'PhysioDirect' telephone assessment and advice services for patients with musculoskeletal problems: economic evaluation

Sandra Hollinghurst,[1] Joanna Coast,[2] John Busby,[1] Annette Bishop,[3] Nadine E Foster,[3] Angelo Franchini,[4] Sean Grove,[5] Jeanette Hall,[5] Cherida Hopper,[1] Surinder Kaur,[1] Alan A Montgomery,[6] Chris Salisbury[1]

▶ Prepublication history this paper is available online. To view these files please visit the journal online (http://dx.doi.org/10.1136/bmjopen-2013-003406).

For numbered affiliations see end of article.

Correspondence to
Sandra Hollinghurst;
s.p.hollinghurst@bristol.ac.uk

## ABSTRACT

**Objectives:** To compare the cost-effectiveness of PhysioDirect with usual physiotherapy care for patients with musculoskeletal problems.

**Design:** (1) Cost-consequences comparing cost to the National Health Service (NHS), to patients, and the value of lost productivity with a range of outcomes. (2) Cost-utility analysis comparing cost to the NHS with Quality-Adjusted Life Years (QALYs).

**Setting:** Four physiotherapy services in England.

**Participants:** Adults (18+) referred by their general practitioner or self-referred for physiotherapy.

**Interventions:** PhysioDirect involved telephone assessment and advice followed by face-to-face care if needed. Usual care patients were placed on a waiting list for face-to-face care.

**Primary and secondary outcomes:** Primary clinical outcome: physical component summary from the SF-36v2 at 6 months. Also included in the cost-consequences: Measure Yourself Medical Outcomes Profile; a Global Improvement Score; response to treatment; patient satisfaction; waiting time. Outcome for the cost-utility analysis: QALYs.

**Results:** 2249 patients took part (1506 PhysioDirect; 743 usual care). (1) Cost-consequences: there was no evidence of a difference between the two groups in the cost of physiotherapy, other NHS services, personal costs or value of time off work. Outcomes were also similar. (2) Cost-utility analysis based on complete cases (n=1272). Total NHS costs, including the cost of physiotherapy were higher in the PhysioDirect group by £19.30 (95% CI −£37.60 to £76.19) and there was a QALY gain of 0.007 (95% CI −0.003 to 0.016). The incremental cost-effectiveness ratio was £2889 and the net monetary benefit at λ=£20 000 was £117 (95% CI −£86 to £310).

**Conclusions:** PhysioDirect may be a cost-effective alternative to usual physiotherapy care, though only with careful management of staff time. Physiotherapists providing the service must be more fully occupied than was possible under trial conditions: consideration should be given to the scale of operation, opening

## ARTICLE SUMMARY

**Strengths and limitations of this study**
- Findings are based on a large sample of patients with a wide range of musculoskeletal problems based in diverse locations.
- The study takes a broad perspective, including the healthcare provider, patients and a valuation of lost productivity.
- Physiotherapists were constrained by trial conditions and were underutilised.

times of the service and flexibility in the methods used to contact patients.

## INTRODUCTION

There is a trend to explore the use of new technology in the delivery of healthcare, particularly the use of telephone assessment and triage as, for example, in National Health Service (NHS) Direct.[1] These services aim to better manage patient demand, and research has shown that telephone-based services can be safe, clinically accurate, cost-effective, acceptable to patients and reduce the workload of clinicians,[2–5] although there have been some concerns about using telephone triage in patients presenting with acute health problems.[6]

Musculoskeletal pain problems are one of the most common causes of disability. Over a quarter of all patients registered in general practice will consult at least once for a musculoskeletal problem each year,[7 8] with musculoskeletal pain accounting for around 15% of all general practitioner (GP) consultations.[9] This high prevalence of musculoskeletal

problems[10] results in large direct and indirect healthcare costs: for example, low back pain alone has recently been estimated to cost the UK economy £15.84 billion a year.[11 12] Most patients are managed with advice and analgesia but many are referred to physiotherapists, with 1.23 million new referrals each year from GPs and 4.4 million in total.[13] Ensuring timely access to physiotherapy has long been an issue within the UK NHS, with waiting times of more than 4 months in some areas. Patients may suffer unnecessary pain and disability, and there are high productivity losses: for example, back pain accounts for some 120 million days of certified absence from work each year.[14] Delay may also cause NHS inefficiencies on the one hand as some patients recover and do not attend their physiotherapy appointment when it finally arrives, while on the other, some patients continue to access more expensive forms of treatment while awaiting their appointment.

In response to these problems, physiotherapy services have drawn on the new service models and a range of 'PhysioDirect' services have been developed. These vary in format though they commonly involve a physiotherapist assessing a patient's musculoskeletal pain problem over the telephone, sometimes supported by computerised assessment templates, offering tailored, self-management advice supplemented by written advice sent by post. Alternatively if the assessment findings suggest that face-to-face care is needed this is arranged, and patients who are initially managed by telephone advice can call back for further advice and/or face-to-face treatment.[15] There is, however, limited evidence about the costs and benefits of this approach within physiotherapy. Local evaluations and non-randomised studies suggest that these services may be popular with patients[16–18] and diagnoses made by physiotherapists over the telephone are comparable to diagnoses reached in face-to-face assessments[19–21] although there is some concern that the experience of the physiotherapist providing the telephone assessment might be important.[19 22]

In particular there is no information about the costs or cost-effectiveness of PhysioDirect services, despite: (1) a major underlying rationale for their development being to generate greater efficiency in the use of resources; and (2) a ready presumption that telephone-based services result in lower costs (by assuming that services better use physiotherapy time, use less costly telephone consultations and reduce rates of appointment non-attendance). Without such evidence, preferably generated alongside high-quality primary evidence obtained using rigorous study designs, it remains unclear whether such services should be more widely implemented. This paper reports the results of an economic evaluation conducted alongside a randomised controlled trial powered to generate evidence on whether PhysioDirect services for primary care patients with musculoskeletal problems produce equivalent outcomes to usual face-to-face services.

## METHOD

### Study design

We conducted an economic evaluation alongside a randomised controlled trial to establish the cost-effectiveness of PhysioDirect compared with usual care based only on face-to-face treatment over a period of 6 months. The trial and its clinical findings have been reported in full elsewhere.[15 23] The aim of the evaluation was to provide information about the long-run costs and benefits of the alternative methods of running a physiotherapy service for this patient group so with that in mind we excluded the initial set up costs associated with establishing the new telephone service, including the training undertaken by the practitioners.[24] As the nature of the intervention suggests there could be an impact on patients' costs, and as it is known that musculoskeletal conditions account for a considerable amount of time off work[14] we chose to use a cost-consequences approach, comparing cost from all three perspectives (healthcare provider, patients and carers, lost productivity) with a range of clinical outcomes.[25 26] However, the perspective of greatest interest to the UK policy makers is that of the health and social care provider[27] so we also conducted a cost-utility analysis to compare cost to the NHS with Quality-Adjusted Life Years (QALYs).

### Setting and participants

We recruited adults aged 18 and over from four community physiotherapy services in England—Bristol, Somerset, Stoke-on-Trent and Cheshire—which provided diversity in terms of socioeconomic status and a mix of urban and rural communities. All patients referred by their GP, or who referred themselves, for physiotherapy for a non-urgent musculoskeletal problem were invited to take part. Patients were randomised to PhysioDirect or usual care on a two to one basis to increase the chances of the PhysioDirect service being fully utilised.

### Interventions

The intervention has been reported in detail elsewhere.[15 23] Patients randomised to the PhysioDirect service received an invitation to telephone a senior (band 6 or above) specially trained physiotherapist, who assessed their musculoskeletal problem with the aid of previously developed computerised templates.[23] These templates were provided by Huntingdonshire Primary Care Trust, which has been operating a similar service since 2001. Patients were then sent appropriate advice leaflets about self-management and exercises to try at home, and invited to phone again and/or make a face-to-face appointment if necessary. If the service was engaged when the patient called, the call was answered by a receptionist who added the patient to a 'call-back' list and the physiotherapist would return the call when they were free. Patients randomised to usual care were put on the usual service waiting list for face-to-face assessment and treatment.

## Outcome measures

We used the EQ-5D-3L[28] valued using the UK tariff,[29] to estimate QALYs gained for the cost-utility analysis. The primary outcome for the trial was the physical component summary (PCS) measure from the SF-36v2 questionnaire[30] and secondary clinical outcomes included: the Measure Yourself Medical Outcomes Profile (MYMOP)[31]; a Global Improvement Score—a single question about overall improvement; a composite measure of response to treatment including pain, function and overall improvement (OMERACT OARSI);[32] patient satisfaction and waiting time to first treatment advice from a physiotherapist. All outcomes (except the global improvement score and waiting time to first treatment advice) were measured at baseline, 6 weeks and 6 months and were obtained from a self-completed questionnaire administered at these three time points.

## Resource use

The analysis was based on costs related to the reason for which the patient was referred to the physiotherapy service. We identified relevant resources in discussion with participating physiotherapists and service managers. Direct costs to the healthcare provider included: cost of initial and follow-up physiotherapy consultations; primary and community consultations; hospital care and prescribed medication. Patient and carer costs included: telephone calls to the PhysioDirect service; travel; over-the-counter medication; prescription costs; private therapy and purchase of equipment; extra domestic help and loss of earnings. Lost productivity was estimated separately in relation to time off work to attend physiotherapy appointments and time off because of the musculoskeletal condition itself.

Patient level data about all physiotherapy appointments and consultations were recorded either automatically by computer or by the physiotherapist treating the patient. For those in the intervention group, the PhysioDirect assessment software recorded which physiotherapist conducted each telephone call, and the duration of each call. In addition to the time logged on to the system physiotherapists had to carry out administrative activities following each telephone call, such as collating information to send to the patient by post. The time spent on these activities was estimated from information available at one site (Bristol) where manual recording of the entire encounter supplemented the electronic recording.

Physiotherapists assigned to the PhysioDirect service were required to be available throughout the time the service was 'open' but they were not usually fully engaged in dealing with patients in the PhysioDirect service during these hours. We conducted an observational time and motion study at each of the four sites to determine how they occupied their non-PhysioDirect contact time in order to apportion costs appropriately. Time and motion data were collected at points in the study when the sites were expected to be fully operational, and across a mix of day, time of day and location. The capital costs required to run a telephone service are potentially less than for a face-to-face service. Each site provided information about space and equipment required to run their telephone service and we used this to estimate an overall percentage reduction of capital costs for these compared with a standard face-to-face service.

Data about all face-to-face appointments were recorded routinely. These data included the length of appointment, the grade of the physiotherapist seen, and information about missed appointments.

Information about other NHS resource use was collected, where possible, from general practice records and supplemented by information gained directly from patients. General practice notes were scrutinised for patient level data on primary care consultations and prescribed medication. We included all consultations at which musculoskeletal condition for which the patient was referred to physiotherapy was mentioned and these were recorded by type of consultation (eg, face-to-face, telephone, out of hours, home visit) and by type of professional seen (eg, GP, nurse). It was not feasible to distinguish between medication prescribed for the condition for which the patient was referred to physiotherapy and any other musculoskeletal problem so we included all medication of a potentially relevant type, defined using British National Formulary (BNF)[33] coding. These were: analgesics (chapters 4.7.1–4.7.2); non-steroidal anti-inflammatory drugs (10.1.1); local corticosteroid injections (10.1.2.2) and drugs for the relief of soft-tissue inflammation (10.3).

A questionnaire was administered to participants at 6 weeks and 6 months after randomisation to obtain resource use data not available elsewhere. The questionnaire was designed specifically for this study but was similar in content and structure to others used for the same purpose.[34] Questions included information about hospital care related to the condition for which the patient was referred to physiotherapy: visits to accident & emergency, outpatient appointments, and inpatient stays. Information about personal expenditure relevant to the patient's musculoskeletal condition was also gained from the questionnaire at 6 weeks and 6 months. We asked about the cost of travel to physiotherapy and other healthcare appointments, expenditure on over-the-counter medication, prescription costs, use of private therapies and their cost, expenditure on equipment or devices and extra help at home. In addition, participants were asked about any time off work, and the associated loss of earnings, because of their condition or to attend healthcare appointments relating to the condition including usual care physiotherapy and PhysioDirect.

## Valuation of resource use

Table 1 gives the unit costs and sources used to value the healthcare resources. We used Curtis[35] to value

**Table 1** Data sources and unit costs

| | Unit cost (£) |
|---|---|
| Primary and community care[35] | |
| General practitioner | |
| Surgery | 27.00 |
| Telephone consultation | 16.00 |
| Home visit | 91.00 |
| Practice nurse | |
| Surgery | 10.00 |
| Telephone consultation | 5.93 |
| Healthcare assistant/phlebotomist | |
| Surgery | 6.92 |
| District nurse | |
| Home visit | 16.33 |
| Out of hours[40 41] | |
| General practitioner | 23.50 |
| Hospital care[36] | |
| A&E | 103.00 |
| Outpatient visits | By Healthcare Resource Group, differentiated by first and follow-up |
| Inpatient stays | By Healthcare Resource Group |
| Prescribed medication[33] | per item, by name, strength and amount |
| Mileage[38] | 0.4612 |
| Time off work[39] | Median national wage by age and sex |

primary and community healthcare and Department of Health reference costs[36] for all hospital-based care. The cost of prescribed medication was estimated from that published in the BNF,[33] adjusted to allow for the discount available to the NHS, and the professional fee and container allowance in accordance with the Drug Tariff for England.[37] Personal expenditure was reported directly by the participants, the exception being travel by car, which was reported as mileage and costed using the AA schedule of motoring costs.[38] Time off work was valued using the median gross weekly earnings by age and sex.[39]

The cost of face-to-face physiotherapy consultations was estimated by adapting the methods of Curtis[35] to obtain a different unit cost for each band of staff at each site. National median pay rates, by band,[42] were adjusted to allow for National Insurance, superannuation, and overheads, as per Curtis, then further adjusted to allow for band and site-specific non-contact time. Information about the proportion of time physiotherapists on each grade typically spend in direct contact with patients was provided by the four physiotherapy service managers. This provided us with a cost per hour for each band of staff at each site.

The unit cost of physiotherapists working in the PhysioDirect service was estimated in a similar way, but allowing for the reduced cost of capital and overheads; information from the site managers indicated this to be about 50%. To obtain a cost per hour of telephone contact we used information from the computerised records of the PhysioDirect service, which identified the proportion of time spent by physiotherapists actually dealing with patients in the PhysioDirect service. We then combined this with data from the time and motion study, which identified activities undertaken during non-contact time, for example, administration relating to face-to-face appointments or general administration, to give a cost per hour for each band of staff at each site.

All costs were valued in £ sterling at 2009 prices, adjusted for inflation where necessary.[35]

### Data analysis

We investigated the amount of each resource used by patients in each group using frequencies, means and medians. Mean total cost per participant was derived by combining resource use with unit costs.

QALYs were derived from responses to the EQ-5D-3 L at baseline, 6 weeks and 6 months using valuations from the UK general population.[29] These values, representing health-related quality of life on a scale between 0 (death) and 1 (best imaginable health), were used to compute QALYs experienced over the 6-month period using the area under the curve approach and adjusting for any difference between the groups at baseline.[43]

A cost-consequences matrix was constructed using all available data. We compared costs from all three perspectives (healthcare provider, patients and carers, lost productivity) with the SF-36v2 PCS, MYMOP, Global Improvement Score, OMERACT OARSI, patient satisfaction, waiting time and QALYs.

The cost-utility analysis was carried out using data on all patients for whom we had complete NHS cost and QALY data. An incremental cost-effectiveness ratio (ICER) was constructed, comparing the difference in mean total cost per patient with mean difference in QALYs, thus the lower the ICER, the greater the cost-effectiveness and the better the value for money.

Uncertainty around the ICER was captured using the bootstrapping technique: 5000 replicates of the cost and QALY data were created by sampling from the original data, with replacement. The range and spread of the 5000 ICERs was used to construct a cost-effectiveness acceptability curve (CEAC) to indicate the likelihood of the intervention being cost-effective. The net monetary benefit (NMB) of the intervention was estimated from the point estimate of the ICER for values of societal willingness-to-pay of £20 000 and £30 000/QALY. If the NMB is positive at a given level of willingness to pay, the intervention is regarded as cost-effective. CIs around the NMB were formed from the bootstrapped estimates.

We used the multiple imputation by chained equation procedure to address the issue of missing cost and EQ-5D data.[44] This technique uses a regression model to estimate missing values from known values. In addition to cost and EQ-5D-3 L variables the imputation model also included randomisation group, age, sex and SF36v2 PCS. Stata V.12[45] was used to generate five datasets using 10 switching procedures.

Discounting was not carried out because the analysis was restricted to costs and outcomes over a period of less than a year. All analyses were conducted using Microsoft Excel and Stata V.12.[45]

### Sensitivity analyses

We addressed three areas of uncertainty using four one/two-way sensitivity analyses. First, we estimated the cost of running the PhysioDirect service if it was operating at full capacity. It is likely that this was not achieved during the trial because of low demand due to exclusions and non-participation in the trial; inflexible staffing levels to ensure consistency throughout the trial period; and the 'one-way' system generally used, where physiotherapists waited for patients to call them but did not routinely contact patients themselves (notwithstanding some limited use of answer-machines). Data from the Bristol service, which continued to operate beyond the trial period and was then able to tailor staffing levels to demand, were used to estimate the cost of running a

more 'efficient but feasible' PhysioDirect service once the trial had ended.

The second area of uncertainty addressed hospital costs. Patients in the trial were recruited from primary care and for these, use of secondary care is infrequent but relatively expensive and this can have a disproportionate effect on mean total cost. We tested this by excluding hospital costs from the total.

The third area of uncertainty tested the effect of using imputed data rather than complete cases; the third sensitivity analysis used trial data with missing values imputed.

Finally, in a two-way sensitivity analysis, we re-estimated the results of the first, (mimicking an 'efficient but feasible' service) in this instance using the imputed dataset.

### RESULTS

A total of 2249 patients were recruited between July and December 2009, and followed up until June 2010, 1506 allocated to PhysioDirect and 743 to usual care. The mean age was 60, with slightly more women than men (60% vs 40%); they were overwhelmingly white (97%), just over half (60%) were employed and all but a few were referred for physiotherapy by their GP. Lower limb problems were the most prevalent (30%) reason for referral, 27% patients had a lumbar problem and 23% upper limb problems. Nearly all participants (2223=99%) gave permission to access their GP notes to obtain data about primary care encounters and prescribed medication. Eighty-one per cent returned questionnaires at both 6 weeks and 6 months though not all participants completed all sections at both time points. We had complete NHS cost and QALY data for 840 (56%) PhysioDirect and 432 (58%) usual care participants.

### Resource use

Table 2 gives information about the different types of physiotherapy consultations by patients in each group. Of the 1506 patients in the PhysioDirect group, 39% (586) were managed solely over the telephone and only

| Table 2 | Number of physiotherapy consultations and mean duration, by type and group | |
|---|---|---|
| | **Usual care (n=743)** | **PhysioDirect (n=1506)** |
| Face-to-face appointments | | |
| Mean (SD) number | 3.11 (2.63) | 1.91 (2.72) |
| Mean (SD) total duration (minutes) | 107.51 (88.92) | 64.20 (89.31) |
| Telephone appointments | | |
| Mean (SD) number | 0.13 (0.44) | 0.96 (0.63) |
| Mean (SD) total duration (minutes) | 4.21 (14.64) | 27.37 (19.92) |
| Home visits | | |
| Mean (SD) number | 0.00 (0.06) | 0.00 (0.06) |
| Mean (SD) total duration (minutes) | 0.14 (2.27) | 0.12 (2.12) |
| All physiotherapy contacts | | |
| Mean (SD) number | 3.25 (2.70) | 2.87 (2.94) |
| Mean (SD) total duration (minutes) | 111.86 (90.50) | 91.70 (95.40) |

**Table 3**  Health services resource use, by group

| | Mean (SD) number of consultations | | | |
|---|---|---|---|---|
| | n | Usual care | n | PhysioDirect |
| GP consultations | 739 | 0.77 (1.47) | 1484 | 0.87 (1.68) |
| Nurse consultations | 739 | 0.04 (0.22) | 1484 | 0.06 (0.32) |
| Other primary care consultations | 739 | 0.02 (0.14) | 1484 | 0.02 (0.17) |
| Total number of primary care contacts | 739 | 0.83 (1.56) | 1484 | 0.96 (1.84) |
| Number of prescriptions | 728 | 1.36 (2.73) | 1469 | 1.68 (3.72) |
| A&E (visits) | 467 | 0.02 (0.01) | 912 | 0.03 (0.01) |
| Outpatient (consultations) | 467 | 0.17 (0.83) | 910 | 0.35 (1.03) |
| Inpatient stays (finished consultant episodes) | 465 | 0.01 (0.10) | 910 | 0.01 (0.10) |
| All available data. | | | | |

46% (695) had any face-to-face consultations. In total, patients in the usual care group had, on average, 0.38 (95% CI 0.12 to 0.63) more consultations than those in the PhysioDirect group and the mean total duration of all consultations was 20 min longer (95% CI 12 to 28).

Tables 3 and 4 give information about NHS and personal resource use. Just over a third of patients (35%) had a GP consultation during the 6 months and 40% received a prescription for musculoskeletal pain-related medication. There was very little difference between the two groups in terms of healthcare use and the only notable difference in personal expenditure was travel to physiotherapy appointments.

## Costs and consequences

Table 5 summarises the mean cost per patient, by group, for each category of cost. All available data are included giving variable denominators for each category. Comparing the two groups, there are small differences in cost in some categories but for most of these the CIs indicate that there is no evidence of a difference between the groups.

Table 6 combines the results of the cost analysis with the full range of primary and secondary outcomes, including QALYs.

Results are presented for all available data, with cost categories combined. Denominators vary within the

table and they also differ from those in table 5 because subcategories have been collapsed. There was no evidence of a difference in the primary clinical outcome (the SF36v2 PCS) between the groups, suggesting that PhysioDirect led to similar outcomes as usual physiotherapy care. Patients in the PhysioDirect group had their first assessment and telephone advice 27 days earlier than those in the usual care group, however patient satisfaction was slightly lower in those receiving PhysioDirect. QALYs were higher in the PhysioDirect group by 0.009, which equates to about 3.3 extra days of full health over a year.

## Cost-utility analysis

The cost-utility analysis presented in table 7 uses complete cases, that is, we include only those patients for whom we had complete NHS cost and QALY data: 432 (58%) from the usual care group and 840 (56%) from PhysioDirect. The small extra cost of caring for patients in the PhysioDirect group was compensated for by the extra QALY gain, giving an ICER of £2889. Values below £20 000 are regarded by National Institute for Health and Care Excellence (NICE) to indicate a cost-effective intervention.[27] At this threshold level of willingness to pay for a QALY there is a positive NMB of £117 (95% CI −£96 to £310) and there is 0.88 probability that the

**Table 4**  Patient and societal resource use, by group

| Number (%) reporting | n | Usual care | n | PhysioDirect |
|---|---|---|---|---|
| Expenditure on travel to physiotherapy | 462 | 242 (52.4) | 1232 | 308 (25.8) |
| Expenditure on travel to primary care | 669 | 116 (17.3) | 1337 | 237 (17.7) |
| Expenditure on over-the-counter medication | 506 | 256 (50.6) | 1028 | 512 (49.8) |
| Expenditure on prescriptions | 559 | 264 (47.2) | 1085 | 508 (46.8) |
| Expenditure on private therapy | 484 | 89 (18.4) | 934 | 167 (17.9) |
| Equipment purchase | 480 | 139 (29.0) | 939 | 233 (24.8) |
| Payments for extra domestic help | 459 | 35 (7.6) | 928 | 76 (8.2) |
| Loss of earnings | 598 | 30 (5.5) | 1209 | 64 (5.3) |
| Any time off to attend physiotherapy consultation | 692 | 218 (31.5) | 1416 | 380 (26.8) |
| Work has been affected because of condition | 477 | 141 (29.6) | 959 | 317 (33.1) |
| All available data. | | | | |

**Table 5** Mean total cost per patient, by group and category. All available data*

| | Usual care | | PhysioDirect | | Incremental difference |
| | n | Mean (SD) cost | n | Mean (SD) cost | (95% CI) |
|---|---|---|---|---|---|
| Physiotherapy services | | | | | |
| Face-to-face appointments | 743 | £64.42 (£53.00) | 1506 | £38.76 (£53.92) | −£25.66 (−£30.37 to −£20.95) |
| Telephone appointments | 743 | £5.22 (£18.01) | 1506 | £35.17 (£26.34) | £29.94 (£27.84 to £32.05) |
| Home visits | 743 | £0.08 (£1.33) | 1506 | £0.08 (£1.46) | £0.00 (−£0.12 to £0.13) |
| Total physiotherapy cost | 743 | £69.73 (£56.17) | 1506 | £74.01 (£63.97) | £4.28 (−£1.12 to £9.69) |
| Primary care services | | | | | |
| GP consultations | 739 | £19.21 (£35.91) | 1484 | £21.69 (£41.66) | £2.48 (−£1.04 to £6.00) |
| Nurse consultations | 739 | £0.44 (£2.37) | 1484 | £0.61 (£3.17) | £0.16 (−£0.10 to £0.42) |
| Other primary care consultations | 739 | £0.03 (£0.57) | 1484 | £0.07 (£1.31) | £0.05 (−£0.05 to £0.15) |
| Total primary care cost | 739 | £19.68 (£36.68) | 1484 | £22.37 (£42.83) | £2.69 (−£0.92 to £6.30) |
| Medication cost | 728 | £11.04 (£51.61) | 1469 | £10.33 (£55.43) | −£0.72 (−£5.53 to £4.10) |
| Hospital services | | | | | |
| A&E | 467 | £1.99 (£17.12) | 912 | £3.17 (£20.84) | £1.18 (−£1.01 to £3.37) |
| Outpatient | 467 | £30.74 (£98.36) | 910 | £38.35 (£126.05) | £7.61 (−£5.50 to £20.72) |
| Inpatient | 465 | £51.02 (£520.48) | 910 | £34.99 (£399.62) | −£16.03 (−£65.70 to £33.64) |
| Total hospital cost | 459 | £83.04 (£561.68) | 899 | £77.00 (£446.24) | −£6.04 (−£60.99 to £48.91) |
| Personal expenditure | | | | | |
| Cost of all calls to physiotherapy service | 743 | £0.97 (£0.99) | 1506 | £1.75 (£1.29) | £0.79 (£0.68 to £0.89) |
| Travel to physiotherapy | 462 | £6.11 (£11.48) | 1232 | £3.11 (£8.51) | −£3.01 (−£4.01 to −£2.00) |
| Travel for primary care | 669 | £0.65 (£2.93) | 1337 | £0.75 (£4.10) | £0.11 (−£0.24 to £0.45) |
| Over-the-counter medication | 490 | £7.67 (£14.09) | 987 | £8.61 (£22.38) | £0.94 (−£1.23 to £3.11) |
| Cost of prescriptions | 553 | £2.72 (£8.95) | 1076 | £2.67 (£8.33) | −£0.05 (−£0.93 to £0.82) |
| Private therapy | 475 | £21.98 (£70.34) | 915 | £39.34 (£296.94) | £17.36 (−£9.76 to £44.48) |
| Equipment purchase | 473 | £17.16 (£169.12) | 924 | £9.12 (£56.42) | −£8.04 (−£20.08 to £4.00) |
| Extra domestic help | 451 | £10.93 (£64.31) | 905 | £13.68 (£96.02) | £2.75 (−£7.07 to £12.56) |
| Cost associated with loss of earnings | 598 | £46.69 (409.72) | 1209 | £82.78 (£885.85) | £36.09 (−£38.63 to £110.81) |
| Value of time off work | | | | | |
| Time off work to attend physiotherapy | 598 | £12.90 (£38.99) | 1211 | £11.91 (£57.86) | £0.95 (−£3.81 to £5.70) |
| Time off work associated with the condition | 452 | £265.92 (£1350.82) | 884 | £226.61(£1139.84) | £111.31 (−£159.04 to £379.67) |

*Uses all available data, so denominators differ by category.

intervention is cost-effective. This is illustrated in the CEAC in figure 1.

**Sensitivity analysis**

The results of the four sensitivity analyses are shown in table 8 and figure 2. Scenario (1) indicates the potential cost-effectiveness of a more efficient PhysioDirect service. During the trial PhysioDirect clinic opening hours, physiotherapists spent about 35% of their time on the phone or dealing with directly related administration; in Bristol after the trial, this was increased to 57%. Under this scenario, the cost per patient in the PhysioDirect group was £14.53 less than under trial conditions and £2.11 less per patient in the usual care group. The ICER is therefore lower at £1045, while the NMB is correspondingly higher at £127 (λ=£20 000). At low levels of λ, the probability of PhysioDirect being cost-effective under this scenario is higher than with the base case, though at λ=£20 000 it reaches a similar value (see figures 1 and 2).

The effect of removing hospital costs from the analysis is shown in sensitivity analysis (2). Hospital costs accounted for 75% of all NHS costs yet only 19% (n=252) participants reported using any secondary care. Hospital use was evenly divided between the two groups so removing these from the analysis made very little difference to incremental analysis.

The effect of imputing missing NHS cost and QALY data is explored in sensitivity analysis (3). Using these data the cost of the interventions is lower but this is offset by higher NHS costs, giving a higher mean total cost in both groups, by £21.41 in the usual care group and £6.58 in the PhysioDirect group. QALYs using imputed data are lower, by 0.005 in the usual care group and by 0.010 in the PhysioDirect group. The net effect is a reduction of both incremental cost and incremental QALYs, giving an ICER of £2260. Uncertainty around the ICER is reduced, as seen by the flatter CEAC in figure 2.

Sensitivity analysis (4) combines analyses (1) and (3) by using imputed cost data in the 'efficient service'

**Table 6** Cost-consequences. All available data*

| | n (%) | Usual care | n (%) | PhysioDirect | Incremental difference (95% CI) |
|---|---|---|---|---|---|
| Mean (SD) cost | | | | | |
| Total physiotherapy cost | 743 | £69.73 (£56.17) | 1506 | £74.01 (£63.97) | £4.28 (−£1.12 to £9.69) |
| Cost of NHS services including physiotherapy | 453 (61%) | £189.19 (£557.61) | 888 (59%) | £196.43 (£472.02) | £7.24 (−£49.68 to £64.10) |
| Total personal expenditure | 310 (42%) | £121.10 (£575) | 714 (47%) | £166.40 (£1040.27) | £45.30 (−£78.01 to £168.61) |
| Total value of all time off work† | 451 (61%) | £276.75 (£1355.00) | 883 (59%) | £240.74 (£1147.20) | −£36.01 (−£174.69 to £102.66) |
| Consequences‡ | | | | | Difference/OR (95% CI)§ |
| SF36v2 PCS | 629 (85%) | 44.18 (10.84) | 1283 (85%) | 43.50 (10.94) | −0.01 (−0.80 to 0.79) |
| MYMOP¶ | 518 (70%) | 2.40 (1.38) | 1033 (69%) | 2.40 (1.43) | −0.02 (−0.16 to 0.11) |
| Global improvement score | 501 (67%) | 4.07 (1.40) | 1001 (66%) | 4.01 (1.44) | −0.08 (−0.23 to 0.08) |
| Response to treatment (OMERACT OARSI) | 510 (69%) | 197 (38.6%) | 1029 (68%) | 430 (41.8%) | 1.14 (0.92 to 1.43) |
| Waiting time to first assessment and advice | 618 (83%) | 34 (20 to 55)** | 1281 (85%) | 7 (4 to 15)** | 0.32 (0.29 to 0.35)†† |
| Patient overall satisfaction | 367 (49%) | 79.7 (26.5) | 739 (49%) | 75.9 (28.3) | −3.8 (−7.3 to −0.3) |
| QALYs‡‡ | 454 (61%) | 0.322 (0.079) | 881 (58%) | 0.331 (0.082) | 0.009 (−0.000 to 0.018) |

*Uses all available data, so denominators differ by category.
†Total of time off to attend physiotherapy and associated with the condition.
‡At 6-month follow-up time point.
§Adjusted for outcome at baseline, gender, age, referral problem, PCT.
¶Lower score is better.
**Median (IQR).
††Accelerated failure time analysis.
‡‡Adjusted for outcome at baseline.
MYMOP, measure yourself medical outcomes profile; NHS, National Health Service; PCS, physical component summary; QALY, Quality-Adjusted Life Year.

**Table 7**  Cost-effectiveness analysis

| | Usual care | | PhysioDirect | | Incremental difference (95% CI) |
|---|---|---|---|---|---|
| | n | Mean (SD) cost | n | Mean (SD) cost | |
| Cost of physiotherapy | 432 | £78.77 (£57.08) | 840 | £86.75 (£65.47) | £7.98 (£0.69 to £15.27) |
| Cost of NHS services other than physiotherapy | 432 | £100.91 (£502.02) | 840 | £112.23 (£476.91) | £11.32 (−£45.08 to £67.72) |
| Total cost including physiotherapy | 432 | £179.68 (£504.73) | 840 | £198.98 (482.12) | £19.30 (−£37.60 to £76.19) |
| QALYs | 432 | 0.325 (0.077) | 840 | 0.332 (0.081) | 0.007 (−0.003 to 0.016) |
| Incremental cost-effectiveness ratio (ICER) | | | | | £2889 |
| Median net monetary benefit (95% CI) based on bootstrapped results | | | | | |
| $\lambda$=£20 000 | | | | | £117 (−£86 to £310) |
| Probability of intervention being cost-effective | | | | | 0.88 |
| $\lambda$=£30 000 | | | | | £184 (−£106 to £461) |
| Probability of intervention being cost-effective | | | | | 0.90 |

Includes cases with complete data on NHS costs and QALYs.
NHS, National Health Service; QALY, Quality-Adjusted Life Year.

scenario. In this case the results indicate that PhysioDirect is, on average, cheaper than usual care with a possible saving of £6.02/patient, which gives a negative value for the ICER, indicating the intervention is superior in terms of both cost and outcome. The probability that the service is cost-effective at $\lambda$=£20 000 is 0.72.

## DISCUSSION
### Statement of principal findings
The results of this economic evaluation suggest that PhysioDirect services for patients with musculoskeletal problems require careful management if they are to be a cost-effective alternative to usual physiotherapy care. There was very little difference between the two groups in terms of either outcomes or costs, and the finding that PhysioDirect is cost-effective is based on evidence that it provides very slightly greater QALY benefits at very slightly greater cost.

Clearer cost savings were observed in the sensitivity analysis that replicated the post-trial service, once greater flexibility in working arrangements was implemented. Without the restrictions of a trial environment staffing was adjusted to meet the anticipated demand, a call-back service was employed which accommodated fluctuations in activity during each session, referrals added to the system were adjusted regularly to reflect actual staffing and the number of patients waiting for a call-back and a higher throughput of patients led to greater economies of scale. These changes ensured physiotherapists within the PhysioDirect service spent a higher proportion of their PhysioDirect clinic time on the telephone with patients. There was no evidence of a difference between PhysioDirect and usual care in cost to patients and their families, or to society through the costs of lost production.

### Strengths and weaknesses of the study
The study has a number of strengths. It is the first study assessing the cost-effectiveness of a PhysioDirect service including a large sample of patients with a wide range of musculoskeletal problems based across a number of

**Figure 1**  Cost-effectiveness acceptability curve (CEAC) showing the probability that the intervention is cost-effective at different levels of willingness to pay for one quality-adjusted life year.

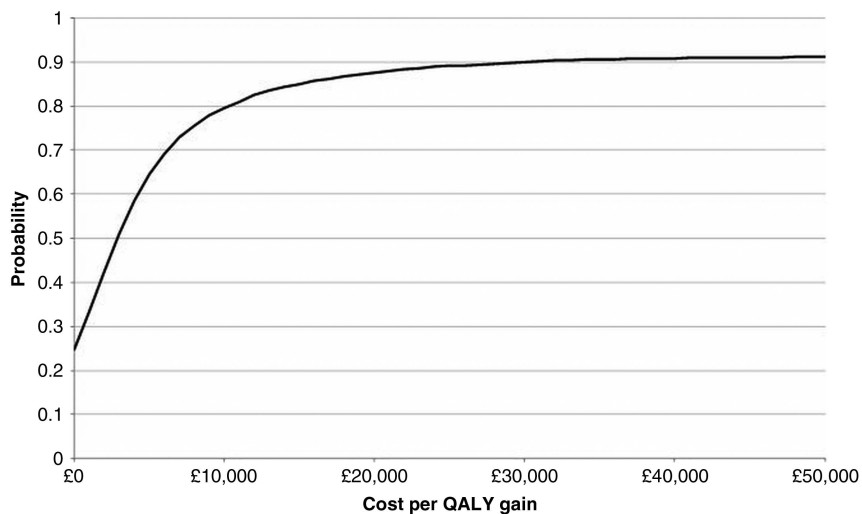

**Table 8**  Sensitivity analysis

| | Usual care | | PhysioDirect | | Incremental difference |
|---|---|---|---|---|---|
| | n | Mean (SD) cost | n | Mean (SD) cost | (95% CI) |
| **(1) Mimicking an efficient service** | | | | | |
| Cost of physiotherapy | 432 | £76.56 (£55.34) | 840 | £72.22 (£61.55) | −£4.34 (−£11.25 to £2.57) |
| Cost of NHS services | 432 | £100.91 (£502.02) | 840 | £112.23 (476.91) | £11.32 (−£45.08 to £67.72 |
| Total cost | 432 | £177.46 (£504.49) | 840 | £184.44 (£481.83) | £6.98 (−£49.89 to £63.85) |
| Incremental cost-effectiveness ratio (ICER) | | | | | £1045 |
| Median net monetary benefit (95% CI) based on bootstrapped results | | | | | |
| λ=£20 000 | | | | | £127 (−£74 to £319) |
| λ=£30 000 | | | | | £193 (−£95 to £473) |
| **(2) Excluding hospital costs** | | | | | |
| Cost of physiotherapy | 448 | £78.49 (£57.14) | 869 | £86.84 (£65.25) | £8.35 (£1.21 to £15.50) |
| Cost of NHS services excluding secondary care | 448 | £33.75 (£76.46) | 869 | £33.49 (£63.76) | −£0.25 (−£8.05 to £7.54) |
| Total cost of NHS services including physiotherapy | 448 | £112.23 (£99.621) | 869 | £120.33 (£98.85) | £8.10 (−£3.21 to £19.41) |
| Incremental cost-effectiveness ratio (ICER) | | | | | £1084 |
| Median net monetary benefit (95% CI) based on bootstrapped results | | | | | |
| λ=£20 000 | | | | | £142 (−£41 to £324) |
| λ=£30 000 | | | | | £217 (−£56 to £489) |
| **(3) Imputed data** | | | | | |
| Cost of physiotherapy | 743 | £69.73 (£56.17) | 1506 | £74.01 (£63.97) | £4.28 (−£1.12 to £9.69) |
| Cost of NHS services | 743 | £131.37 (£465.49) | 1506 | £131.51 (£384.36) | £0.17 (−£36.13 to £36.48) |
| Total cost of NHS services including physiotherapy | 743 | £201.09 (£467.51) | 1506 | £205.55 (£390.04) | £4.46 (−£32.22 to £41.14) |
| Quality-adjusted life years | 743 | 0.320 (0.003) | 1506 | 0.322 (0.002) | 0.002 (−0.006 to 0.009) |
| Incremental cost-effectiveness ratio (ICER) | | | | | £2260 |
| Median net monetary benefit (95% CI) based on bootstrapped results | | | | | |
| λ=£20 000 | | | | | £34 (−£119 to £193) |
| λ=£30 000 | | | | | £52 (−£172 to £285) |
| **(4) Imputed data and 'efficient' service** | | | | | |
| Cost of physiotherapy | 743 | £67.61 (£54.19) | 1506 | £61.41 (£59.13 | −£6.20 (−£11.26 to −£1.14) |
| Cost of NHS services | 743 | £131.37 (£465.49) | 1506 | £131.54 (384.36) | £0.17 (−£36.13 to £36.48) |
| Total cost of NHS services including physiotherapy | 743 | £198.98 (£467.48) | 1506 | £192.95 (£389.52) | −£6.02 (−£42.68 to £30.63) |
| Quality-adjusted life years | 743 | 0.320 (0.003) | 1506 | 0.322(0.002) | 0.002 (−0.006 to 0.009) |
| Incremental cost-effectiveness ratio (ICER) | | | | | −£3054 |
| Median net monetary benefit (95% CI) based on bootstrapped results | | | | | |
| λ=£20 000 | | | | | £47 (−£113 to £202) |
| λ=£30 000 | | | | | £67 (−£165 to £293) |

NHS, National Health Service.

locations.[15] It uses a rigorous study design and conforms to CONSORT guidelines.[46] The follow-up rate from participating patients was in line with other primary care trials,[47] [48] a high proportion of the resource use data were collected from GP records, and there was collection of resource use information outside of the main health service perspective. The cost consequences analysis provides complete information on costs from different perspectives compared with a range of outcomes so although this approach is sometimes criticised for leaving the reader to evaluate the findings it does have the advantage of transparency. Furthermore, in this study we have also presented a cost-utility analysis that conforms to the recommendations of NICE. Nevertheless, there are also limitations. The practices recruited to the trial had a low proportion of ethnic

minority patients, a slightly lower proportion of patients from deprived areas were judged to be eligible, and the proportion of eligible individuals consenting to participate in the trial was only 50%.[23] These factors limit the generalisability of the results though none of these selection effects was large. Further, a particular difficulty in conducting economic evaluation with new service developments is ensuring that they are fully utilised,[24] particularly when conducting analysis from a long-run perspective, as here. Although there was a clear run-in period prior to data collection for the trial (ranging from 4 to 12 weeks in each of the four sites), to ensure that services were operating as well as they could, and a 2:1 randomisation ratio in favour of PhysioDirect was used, there was still considerable underutilisation of the new service. This was ameliorated by including a more

**Figure 2** Cost-effectiveness acceptability curve (CEAC) showing the probability that the intervention is cost-effective at different levels of willingness to pay for one quality-adjusted life year: sensitivity analyses.

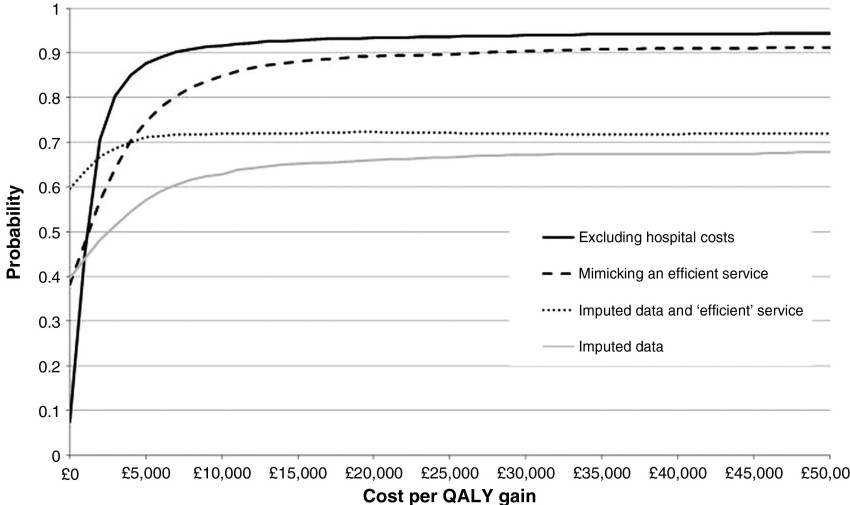

fully utilised service within the sensitivity analysis. It should also be noted that, because of the use of a long-run perspective in the analysis, set up costs, which may, in the short-term be important in a financially constrained service, are not included here. Finally, because the differences in both costs and effects are small, there is still some uncertainty around the findings.

### Meaning of the study and implications for policymakers

If the aim of health services is to achieve maximum health gain from an investment in healthcare, then PhysioDirect has a high probability of being more cost-effective than usual physiotherapy care. This research also, however, suggests that it cannot be assumed that PhysioDirect will reduce costs, although it could potentially do so if PhysioDirect services are managed efficiently. Both the costs and cost-effectiveness of services will depend on the productivity of physiotherapist time. If physiotherapists are able to use most of their time dealing directly with patients during sessions when they are available on the telephone, then the service will be less costly. This is most likely to be achieved by operating a call-back service and/or by operating the service on a large scale to even out fluctuations in demand. A larger scale operation might also offer other economies of scale in terms of infrastructure requirements. Further efficiencies may also be achieved if these services are, in the future, provided in conjunction with direct access for patients (rather than following referral from another healthcare professional), given that patients who self-refer are likely to contact the service with musculoskeletal problems of shorter duration,[49] and such patients may be particularly appropriate for the initial assessment and advice provided by a PhysioDirect service. More generally, the study has broader implications for telephone services, particularly around the implicit assumption that such services will inevitably be money saving. Here this assumption was found to be false, largely because the physiotherapists' time was underutilised during PhysioDirect clinic hours. Thus, for all such

services, it will be important for policymakers to ensure that easy assumptions about the costs of these services are properly assessed in relation to factors such as how efficiently the service is run, and what proportion of patients are subsequently invited for face-to-face care following an initial telephone call.

### Unanswered questions and future research

As services evolve, further research should explore the costs and benefits of PhysioDirect under different scenarios. These might include: comparing different skill levels of staff operating the service; the inclusion of patient self-referrals in addition to GP referrals; the use, or otherwise, of computerised support in assessing the patient; the extension to internet services (possibly combined with cameras);[50] and the use of mobile 'smartphone' technology, for example, in rapid assessment of musculoskeletal injuries. In particular, however, it will be important to assess the costs and benefits of services once they are more established and provided on a wider scale. The costs and benefits of telehealth more generally need further exploration in relation to their cost effectiveness particularly given the negative findings of the Whole Systems Demonstrator project evaluating telehealth support and treatment for patients with long-term conditions.[51] It would be helpful to identify those characteristics that are likely to make services both more cost-effective and less costly.

**Author affiliations**
[1]Centre for Academic Primary Care, School of Social and Community Medicine, University of Bristol, Bristol, UK
[2]Health Economics Unit, School of Health & Population Sciences, University of Birmingham, Birmingham, UK
[3]Arthritis Research UK Primary Care Centre, Primary Care Sciences, Keele University, Keele, UK
[4]Imperial Clinical Trials Unit, School of Public Health Medicine, Imperial College, London, UK
[5]Musculoskeletal Outpatient Department, Bristol Community Health, Bristol, UK
[6]Nottingham Clinical Trials Unit, Queen's Medical Centre, Nottingham, UK

**Correction notice** This article has been corrected since it was published Online First. The Open Access statement has been corrected.

**Acknowledgements** The authors would like to dedicate this paper to the memory of Cherida Hopper, trial manager of the PhysioDirect study, who sadly died in June 2013. We would like to thank the patients who contributed to this research; the physiotherapists, administrative staff, managers and commissioners who supported the set up and delivery of the trial in the four primary care trusts; participating general practices; the research support staff in Bristol and Keele; members of the Trial Steering Committee and Data Monitoring Committee; and Jill Gamlin and Nick Deane who developed the PhysioDirect assessment algorithms and software used in this trial.

**Contributors** CS was principal investigator on the trial, SH took the lead in the economic evaluation. JC, NEF, SG, JH and AAM were coapplicants and contributed to the conception and design. SH, JC and JB carried out the analysis with help from CS and AAM, and all authors contributed to the interpretation of the data. SH and JC wrote the first draft of the paper with all authors contributing to subsequent revisions. All authors have seen and approved the final version.

**Funding** The research was funded by the Medical Research Council (MRC) and managed by the National Institute for Health Research (NIHR) on behalf of the MRC-NIHR partnership. The funder had no role in: the study design; the collection, analysis or interpretation of data; the writing of the report; the decision to submit the paper for publication. The researchers are all independent of the funders. All researchers had access to all the data. Nadine Foster is supported by a National Institute for Health Research NIHR Research Professorship NIHR-RP-011-015. The views expressed in this publication are those of the authors and not necessarily those of the NHS, the National Institute for Health Research or the Department of Health.

**Competing interests** None.

**Ethics approval** Multisite research ethics approval was obtained from Southmead research ethics committee, reference 08/H0102/95.

**Provenance and peer review** Not commissioned; externally peer reviewed.

**Data sharing statement** Participants did not give informed consent for data sharing but the data are anonymised and the risk of identification is low. Data from the trial may be available from the corresponding author subject to agreement about the use of the data.

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
