## [Reviewer comments · BMJ Open]

Some articles will have been accepted based in part or entirely on reviews undertaken for other BMJ Group journals. These will be reproduced where possible.

ARTICLE DETAILS

TITLE (PROVISIONAL)	A pragmatic randomised controlled trial of 'PhysioDirect' telephone assessment and advice services for patients with musculoskeletal problems: economic evaluation
AUTHORS	inghurst, Sandra; Coast, Joanna; Busby, John; Bishop, Annette; Foster, Nadine; Franchini, Angelo; Grove, Sean; Hall, Jeanette; Hopper, Cherida; Kaur, Surinder; Montgomery, Alan; Salisbury, Chris

VERSION 1 - REVIEW

REVIEWER	Ross Iles PhD, Dip Work Disability Prevention, B Physio (Hons) Senior Lecturer Department of Physiotherapy Monash University Australia. No competing interests.
REVIEW RETURNED	17-Jul-2013

THE STUDY	A minor point regarding the references used to describe costs/epidemiology of LBP are over 10 years old. A recent SR by Balagun in Lancet would be a stronger source. Please note that the statistical methods used are outside my field of expertise. I recommend a referee with experience in this field of analysis ensures the methods are appropriate. On surface value they seem appropriate. In the list above I have noted the main outcome measures are not clear. I believe they are clear for people who have read and are experienced in the area, but require clearer definition for the typical reader. This would be especially beneficial in describing what outcomes are more desirable in terms of ICER and net monetary benefit.
RESULTS & CONCLUSIONS	While I understand it is difficult to compare the results to other telehealthcare interventions, previous evidence could be drawn upon to a greater extent in the discussion.
REPORTING & ETHICS	This article is very similar to the content of the following report: Salisbury C, Foster NE, Hopper C, Bishop A, Hollinghurst S, Coast J, et al. A pragmatic randomised controlled trial of the effectiveness and cost-effectiveness of 'PhysioDirect' telephone assessment and advice services for physiotherapy. Health Technol Assess 2013;17. However, the original RCT upon which this economic analysis is based is also part of this report and appears in BMJ. This report is

	also one of the references in the article so I conclude this is not an issue.
GENERAL COMMENTS	I would like to congratulate the authors for the apparent thoroughness of their economic evaluation. I say apparent because economic evaluation is not an area of expertise of mine, so my approach to the review of this article was to ensure it could be followed by a reader without experience in reading this type of article. To that end the authors have done an excellent job in clearly describing all attempts to gather all relevant cost data. I do believe there are aspects of the article that could be made clearer to the reader to guide them through the interpretation of the data (please see attached for detail). The findings of the article are of particular importance, as noted in the discussion, that interventions of this type should not be assumed to be cost effective. Just like other interventions, therapist time must be used efficiently with good systems in place. This is an important article and an important message to get "out there". Please find attached my comments as I read through the article. My recommendation is to accept the article rather than minor amendments because my comments on the whole are minor and the article could be published as is - however I encourage the authors to address these comments as I believe they will improve the manuscript and would serve to clarify the study for the readers.

REVIEWER	Dr Sarah Wordsworth Senior Researcher Health Economics Research Centre Nuffield Department of Population Health Rosemary Rue Building Old Road Campus Headington University of Oxford
REVIEW RETURNED	22-Jul-2013

THE STUDY	I have written no to whether 'the patients representative of actual patients the evidence might affect?'. This is because I am unsure about the generalisability of the results from the actual trial. In particular, I would like to see information in the paper commenting on whether the trial centres were representative of the patient population. 97% white is very high, and I would have liked some comments on this also. Also no comment is made on the result that the trial had more females than males. I think that the main outcome for the economic evaluation is abit confusing given that there are different outcomes measures being used. It could be helpful to guide the reader more.
GENERAL COMMENTS	This paper is very well written and comprehensive. There are just a few points of clarification/areas where extra information could be useful. 1. Could the authors please justify more fully why they used 2 different forms of economic evaluation, i.e. a cost-consequence analysis and a cost-utility analysis. Readers will be more familiar with just one form of economic evaluation being used at a time, generally cost-utility analysis. Whilst lots of interesting information is reported in the paper, it is maybe abit confusing in parts having 2

	different approaches. Also, I think that it would be useful to make some comment about the potential limitations of CCA. In particular, that it leaves the reader to judge the most important outcome and that it maybe doesn't address uncertainty in the data in a systematic way. I personally think CCA is very useful, but am aware that this view isn't shared amongst some health economists. So I think that more early on justification for using it would help. 2. More information on data analysis for the reader would be useful, especially describing the bootstrapping and what is meant by net benefit. 3. I am unclear as to why the authors used imputation for missing costs and QALYs in the sensitivity analysis, as most health economists tend to use imputation methods in their base case analysis. Also, explaining that 2 different approaches to handling the missing data (complete case analysis and imputation), takes up a great deal of space in the paper which could have been used for other information. 4. Was it the case that patients made the initial phone call to PhysioDirect and then staff called them back? This is unclear in the paper. 5. It would be useful to report the range of musculoskeletal conditions in the paper and say whether there was any difference in costs and outcomes across different conditions. 6. For resource use - Could the authors make it clear whether they produced a bespoke questionnaire to collect information such as travel, loss of earnings etc? Could they make the questionnaire available to readers on request? 7. How were staff chosen to be included in the trial (aside from staff grade)? Also did they have special training to perform the telephone consultations and was this costed? 8. Why are costs reported in 2009 prices? The main trial was published in 2013, so I would have thought the authors would inflate the figures to maybe 2012. 9. Table 5. The heading is missing for the PhysioDirect column.
--	--

VERSION 1 – AUTHOR RESPONSE

Reviewer: Ross Iles
 PhD, Dip Work Disability Prevention, B Physio (Hons)
 Senior Lecturer
 Department of Physiotherapy
 Monash University
 Australia.
 No competing interests.

A minor point regarding the references used to describe costs/epidemiology of LBP are over 10 years old. A recent SR by Balague in Lancet would be a stronger source.
 Thank you, we have now replaced reference 10 with this reference.

Please note that the statistical methods used are outside my field of expertise. I recommend a referee with experience in this field of analysis ensures the methods are appropriate. On surface value they seem appropriate.

In the list above I have noted the main outcome measures are not clear. I believe they are clear for people who have read and are experienced in the area, but require clearer definition for the typical reader. This would be especially beneficial in describing what outcomes are more desirable in terms

of ICER and net monetary benefit.

We have expanded the relevant paragraph in method/data analysis to describe more fully how ICERs are formed, how they are interpreted, what net monetary benefit is, and how the bootstrapping technique is used to capture uncertainty around the point estimates.

While I understand it is difficult to compare the results to other telehealthcare interventions, previous evidence could be drawn upon to a greater extent in the discussion.

We have not been able to find any studies similar enough to this one to be able to draw any interesting conclusions about comparability. We note the recently published results of the Whole Systems Demonstrator project (ref #51) and remark on the need for more research in this area.

This article is very similar to the content of the following report:

Salisbury C, Foster NE, Hopper C, Bishop A, Hollinghurst S, Coast J, et al. A pragmatic randomised controlled trial of the effectiveness and cost-effectiveness of 'PhysioDirect' telephone assessment and advice services for physiotherapy. *Health Technol Assess* 2013;17.

However, the original RCT upon which this economic analysis is based is also part of this report and appears in *BMJ*. This report is also one of the references in the article so I conclude this is not an issue.

I would like to congratulate the authors for the apparent thoroughness of their economic evaluation. I say apparent because economic evaluation is not an area of expertise of mine, so my approach to the review of this article was to ensure it could be followed by a reader without experience in reading this type of article. To that end the authors have done an excellent job in clearly describing all attempts to gather all relevant cost data. I do believe there are aspects of the article that could be made clearer to the reader to guide them through the interpretation of the data (please see attached for detail).

The findings of the article are of particular importance, as noted in the discussion, that interventions of this type should not be assumed to be cost effective. Just like other interventions, therapist time must be used efficiently with good systems in place. This is an important article and an important message to get "out there".

Please find below my comments as I read through the article. My recommendation is to accept the article rather than minor amendments because my comments on the whole are minor and the article could be published as is - however I encourage the authors to address these comments as I believe they will improve the manuscript and would serve to clarify the study for the readers.

Abstract

Need to state in the abstract that the differences were not significant (don't leave it up to the reader to evaluate the 95% CI).

Throughout the paper we have presented 95% confidence intervals to indicate uncertainty around our estimates. Use of the term 'significant' places undue emphasis on an arbitrary threshold (see Sterne and Davey-Smith <http://www.bmj.com/content/322/7280/226.1>) therefore we have not used this in the abstract as suggested by the reviewer, nor anywhere else in the paper.

Key messages – nice summary

Note similarity with Salisbury C, Foster NE, Hopper C, Bishop A, Hollinghurst S, Coast J, et al. A pragmatic randomised controlled trial of the effectiveness and cost-effectiveness of 'PhysioDirect' telephone assessment and advice services for physiotherapy. *Health Technol Assess* 2013;17. I'm

not sure where this sits because the study itself reports that the original RCT was reported elsewhere and cites the full report in Health Technology Assessment. I assume this is ok since the original RCT also appears in BMJ (so a similar situation with the economic analysis).

References regarding costs associated with LBP are over 10 years old.
We have replaced references 10 and 11 with more recent ones.

Was the specialised training taken into account in the economic analysis?

We have not included the cost of training the physiotherapists in the analysis, neither have we included the cost of equipment required to set up the PhysioDirect service as we have focused on the long-run costs as is usual practice in economic evaluation. We have now made this clear in the method/study design section, inserted a reference to justify this decision and discussed the consequences of this in the discussion/the meaning of the study and implications for policy makers section.

Page 9 line 16 – minor grammatical area. Delete unnecessary “a”
Thank you, this has been corrected by deleting the ‘s’ from “telephone services”.

The method used to gather other cost data – is this typically gathered via questionnaire? In economic analysis literature are there sources to support this approach? Methods seem comprehensive but this type of analysis is not an area of strength. Other sources and valuations are well supported – can this section be too?

This is a common method, used in the majority of patient-level economic evaluations. To make this clear we have included, in the methods/resource use section, a reference to the website of a compilation of similar questionnaires used in other studies (www.dirum.org)

Table 1

Interesting that a GP out of hours is a lower cost than the GP at the surgery? Is this correct?

Reliable and comparative data on the cost of out-of-hours services are very difficult to come by. For this analysis we used two recent publications (refs #41 and #42) to guide us. Although it may seem odd that an OOH consultation is estimated to be less expensive than one in a surgery, this is because many OOH consultations are conducted over the telephone and are generally shorter in length than surgery consultations.

Page 12 lines 16-21. This analysis lies outside my area of expertise. Please check with someone experienced in these methods to determine whether this approach is appropriate.

We have expanded this paragraph to make the method more transparent (see previous response)

Page 13 Sensitivity analysis – opening statement says addressed three areas of uncertainty but four analyses are described in this section. Re-word this section as the final paragraph is not necessary – it belongs with the first description.

Three areas of uncertainty were addressed in four analyses. We have re-drafted this paragraph to make this clearer.

Page 13 line 19: delete unnecessary comma.

Done

Results

Page 14 – is it common to have complete data for 60% in these types of analyses? It would be good to know if this is a low, usual or high proportion compared to other studies of this type. This is mentioned later in the article but can references be provided?

This level of completeness is reasonable in a primary care trial (see references #47 and #48 inserted in discussion/strengths and weaknesses). The total percentage of missing observations was far less than 60% because 99% participants gave permission to access their notes and 81% provided some questionnaire information, however these analyses only included the 60% of people with complete data on all relevant variables. Resources used in delivering physiotherapy was complete for all patients in both groups.

Page 14 line 24: What do you mean by “only important difference”? Was this statistically significant, perhaps clinically significant? This statement needs to be qualified for the reader.

We did not perform any statistical tests on these differences and the comment refers to simple ‘eyeballing’ of the proportions in each arm. We have altered to “notable difference” in the text.

Table 5: I think it would be useful to indicate to the reader which differences are significant, rather than leaving it to the reader to interpret a table with a large number of confidence intervals. I would prefer to indicate the significant results and then allow the reader to determine the importance of the difference in cost, rather than requiring the reader to interpret both. I suggest a superscript or underline to indicate significant differences.

Please refer to the response above about statistical significance.

Table 6: given QALYs are the primary outcome for the cost analysis, I suggest highlighting this for the reader (e.g. bold) and stating that higher numbers are better. It is in the text, but I think guiding the reader to the primary outcome in a table such as this is a good idea.

In a cost-consequences framework no outcome is more important than any other. All are presented equally for the reader to make judgements depending on their own interests and circumstances. We have referred to the following two papers in the method/study design section to reinforce this: Coast, J. "Is economic evaluation in touch with society's health values?" *BMJ* 2004; (329): 1233-1236 and Drummond M. Use of Pharmacoeconomics Information—Report of the ISPOR Task Force on Use of Pharmacoeconomic/Health

Economic Information in Health-Care Decision Making. *Value in Health* 2003; 6(4)

Page 15 line 16: Delete unnecessary “again”.

Done

Page 15 final paragraph. I have a problem with the text referring to a statistically significant result (lower satisfaction) in the same manner as a non-significant result (QALYs – albeit very close to significance). In fact, the writing refers to a slightly lower satisfaction result – but this was a clearer difference than the “higher QALYs”. Make sure you don’t lead the reader with descriptions of the results.

Please refer to the response above about statistical significance. Regarding the point about direction, we do include a footnote to table 6 explaining that contrary to the other outcomes, lower scores are

better in the case of MYMOP.

Help the reader interpret the ICER by stating it is the cost per QALY gained – and that a lower ICER is desirable (not all readers will be able to interpret this statistic without guidance).

We have made this clearer in the method/data analysis section and reiterated here in the result section.

The same goes for the net monetary benefit. Please provide plain language explanations for these terms, how they are determined and a suggestion for how they should be interpreted.

We have made this clearer in the method/data analysis section and reiterated here in the result section.

Table 8: formatting needs adjusting so all figures are aligned – it becomes hard to follow the table. Yes, we have re-aligned the table now

Please explain the significance of the negative ICER in the 4th analysis in Table 8. The text reports different figures to those in other sensitivity analyses, meaning the reader is not left with a clear interpretation of the analysis.

We have explained that the negative ICER is a result of the cost saving under this scenario - in contrast to the other sensitivity analyses - and that this represents a cheaper and more effective intervention.

Discussion

Good discussion – it covered my impressions of the study nicely and pointed out some good directions for future research. There could be greater reference to other research in this section, but I understand this is difficult given the fact this is the first economic evaluation of this type.

We have not been able to find any studies similar enough to this one to be able to draw any interesting conclusions about comparability. We note the recently published results of the Whole Systems Demonstrator project (ref #51) and remark on the need for more research in this area.

Reviewer: Dr Sarah Wordsworth
Senior Researcher

Health Economics Research Centre
Nuffield Department of Population Health
Rosemary Rue Building
Old Road Campus
Headington
University of Oxford
OX3 7AE

I have written no to whether 'the patients representative of actual patients the evidence might affect?'. This is because I am unsure about the generalisability of the results from the actual trial. In particular, I would like to see information in the paper commenting on whether the trial centres were

representative of the patient population. 97% white is very high, and I would have liked some comments on this also. Also no comment is made on the result that the trial had more females than males.

The trial population was largely representative of the population served by the practices recruited, which did not include areas with a high proportion of patients from non-white ethnic backgrounds. Even so, we agree that the proportion of white participants is surprisingly high, and we do not know if this suggests that people from other ethnic groups are less likely to be referred for physiotherapy or to participate in the research, since we only have data about ethnicity in people who agreed to participate. With regard to gender, the higher proportion of females in the trial reflects the fact that 59% of those referred to physiotherapy were female. We have inserted a comment on generalisability in the discussion/strengths and weaknesses of the study section.

I think that the main outcome for the economic evaluation is a bit confusing given that there are different outcomes measures being used. It could be helpful to guide the reader more.

It is unclear whether the comment about “main outcome for the economic evaluation” refers to the main result or the primary outcome. However, we have slightly reworded the section in the abstract about primary and secondary outcomes to make it clear that all were included in the cost consequences framework.

This paper is very well written and comprehensive. There are just a few points of clarification/areas where extra information could be useful.

1. Could the authors please justify more fully why they used 2 different forms of economic evaluation, i.e. a cost-consequence analysis and a cost-utility analysis. Readers will be more familiar with just one form of economic evaluation being used at a time, generally cost-utility analysis. Whilst lots of interesting information is reported in the paper, it is maybe a bit confusing in parts having 2 different approaches. Also, I think that it would be useful to make some comment about the potential limitations of CCA. In particular, that it leaves the reader to judge the most important outcome and that it maybe doesn't address uncertainty in the data in a systematic way. I personally think CCA is very useful, but am aware that this view isn't shared amongst some health economists. So I think that more early on justification for using it would help.

We have addressed this by re-ordering and extending the paragraph method/study design. We have also inserted a reference to support the use of cost consequences for this study. In the discussion/strengths and weaknesses section we have added a few sentences to justify using both cost consequences and cost utility analysis.

2. More information on data analysis for the reader would be useful, especially describing the bootstrapping and what is meant by net benefit.

Yes, this has been addressed (as per reviewer 1)

3. I am unclear as to why the authors used imputation for missing costs and QALYs in the sensitivity analysis, as most health economists tend to use imputation methods in their base case analysis. Also, explaining that 2 different approaches to handling the missing data (complete case analysis and imputation), takes up a great deal of space in the paper which could have been used for other information.

We are not aware of a 'usual practice' regarding imputation of missing data in economic evaluation. A recent review (Noble S et al “Missing data in trial-based cost-effectiveness analysis: the current state of play” *Health Economics* 2012; 21: 187–200) was inconclusive on this point, which is a pity because it would be useful to have guidance. Whether complete cases or imputed data are used in the base

case findings, it seems good practice to present the alternative in a sensitivity analysis - so the decision as to which way round to do it is to some extent arbitrary. We are comfortable with our decision.

4. Was it the case that patients made the initial phone call to PhysioDirect and then staff called them back? This is unclear in the paper.

Patients were invited to call the PhysioDirect service but if all the physiotherapists were engaged, the call would be answered by a receptionist who would take the details and add them to a list of patients waiting to be called back. We have now explained this.

5. It would be useful to report the range of musculoskeletal conditions in the paper and say whether there was any difference in costs and outcomes across different conditions.

We have inserted information about the most common musculoskeletal problems in the results section. The question of costs and outcomes across the different conditions is interesting but is beyond the scope of this paper. We have investigated this with regard to the primary clinical outcomes but power for this kind of sub-group analysis is limited and no differences between sub-groups based on type of musculoskeletal problem were found. Results are available in the full HTA report.

6. For resource use - Could the authors make it clear whether they produced a bespoke questionnaire to collect information such as travel, loss of earnings etc? Could they make the questionnaire available to readers on request?

Yes, we have now made that clear (see reviewer 1). Very happy to make the questionnaire available, it could be published as an appendix.

7. How were staff chosen to be included in the trial (aside from staff grade)? Also did they have special training to perform the telephone consultations and was this costed?

Grade of staff was the main criterion for identifying staff to work on the trial. The actual selection was handled pragmatically by the service manager at each site and could have varied depending on individual circumstances. The staff did have special training which is explained in the related trial paper and the HTA report. We have now mentioned the training in the revised paper and explained why we did not include this in the economic analysis.

8. Why are costs reported in 2009 prices? The main trial was published in 2013, so I would have thought the authors would inflate the figures to maybe 2012.

Patients were recruited between July and December 2009 and followed up until June 2010. When analysis began the most up to date unit costs data were for 2009.

9. Table 5. The heading is missing for the PhysioDirect column.

Thank you, that has now been inserted.